# Genome-Wide Comparison of Structural Variations and Transposon Alterations in Soybean Cultivars Induced by Spaceflight

**DOI:** 10.3390/ijms232213721

**Published:** 2022-11-08

**Authors:** Hangxia Jin, Xujun Fu, Xiaomin Yu, Longming Zhu, Qinghua Yang, Fengjie Yuan

**Affiliations:** 1Hangzhou Sub-Center of National Soybean Improvement, Institute of Crop and Nuclear Technology Utilization, Zhejiang Academy of Agricultural Sciences, Hangzhou 310021, China; 2Zhejiang Key Laboratory of Digital Dry Land Crops, Zhejiang Academy of Agricultural Sciences, Hangzhou 310021, China

**Keywords:** spaceflight, soybean, structural variations, transposable elements

## Abstract

Space mutation causes genetic and phenotypic changes in biological materials. Transposon activation is an adaptive mechanism for organisms to cope with changes in the external environment, such as space mutation. Although transposon alterations have been widely reported in diverse plant species, few studies have assessed the global transposon alterations in plants exposed to the space environment. In this study, for the first time, the effects of transposon alterations in soybean caused by space mutation were considered. A new vegetable soybean variety, ‘Zhexian 9’ (Z9), derived from space mutation treatment of ‘Taiwan 75’ (T75), was genetically analyzed. Comparative analyses of these two soybean genomes uncovered surprising structural differences, especially with respect to translocation breakends, deletions, and inversions. In total, 12,028 structural variations (SVs) and 29,063 transposable elements (TEs) between T75 and Z9 were detected. In addition, 1336 potential genes were variable between T75 and Z9 in terms of SVs and TEs. These differential genes were enriched in functions such as defense response, cell wall-related processes, epigenetics, auxin metabolism and transport, signal transduction, and especially methylation, which implied that regulation of epigenetic mechanisms and TE activity are important in the space environment. These results are helpful for understanding the role of TEs in response to the space environment and provide a theoretical basis for the selection of wild plant materials suitable for space breeding.

## 1. Introduction

Soybean (*Glycine max* (L.) Merr.) is an important oil crop and a high-protein food crop widely grown in China and elsewhere in the world. Soybean is also a popular vegetable in China and East Asia. To develop new varieties of elite vegetable soybean and to meet the growing market demand, our research group selected a new vegetable soybean variety, ‘Zhexian 9’ (Z9), by a pedigree breeding method from the wild type ‘Taiwan 75’ (T75), using space mutation treatment facilitated by the recoverable satellite Shijian 8. Shijian 8, the first dedicated breeding satellite in China, was launched at 15:00 on 9 September 2006, at the Jiuquan Satellite Launch Center. The satellite was in orbit for 15 days. The satellite recovery capsule landed in central Sichuan Province at 10:43 on 24 September. The perigee altitude of satellite operation was 180 km, the apogee altitude was 469 km, and the orbital inclination was 63°. During satellite operation, the temperature in the recovery cabin ranged between 20.72 and 7.21 °C. The radiation dose to the sample-carrying part of the satellite module was 2.894 mGy and the average gravity was 1.3 × 10^−3^ g [1,2]. The T75 seeds were affected by the synergy of space radiation, space microgravity, and other space-related factors. The variety Z9 was selected using a pedigree breeding method for 6 years. Z9 exhibits superiority in a broad array of agronomic traits, including yield, good eating quality, and resistance to soybean mosaic virus and salt stress [3,4]. However, the progeny of Z9 are more susceptible to phenotypic variation than T75.

Space mutation is a technology that makes use of space environmental factors to cause changes in the genetic characteristics of biological materials, which are then screened and cultivated to potentially select new strains or varieties. The space environmental factors include cosmic irradiation, microgravity, and space magnetic fields, which might induce DNA breakage, recombination, genomic instability, and transposon activation [5,6,7]. A hypothesis was formulated that transposable elements (TEs) are involved in transgenerational transmission and accumulation of mutations by the offspring of space-mutated materials. The TEs might promote genomic restructuring and plasticity to generate genetic, biochemical, and phenotypic novelties to adapt to external changes, such as the space environment, pathogens, wounding, temperature, drought, or X-ray irradiation [8,9,10]. The transposons in plant genomes are extremely rich and diverse and have played an important role in genome structure, function, and evolution. The first TE nomenclature organized TEs into two classes according to their transposition intermediates in 1989 [11]: (1) an RNA intermediate for retrotransposons (class I elements); and (2) a DNA intermediate for DNA transposons (class II elements). This classification has been further refined since 2007, which splits TEs into the two cited classes (Class I and Class II), then into subclasses, orders, and superfamilies.

Mounting evidence indicates that TEs constitute the genomic fraction that is susceptible and responsive to environmental perturbations and hence most likely to manifest genetic instabilities in times of stress [12,13,14,15,16]. It was further established that a major reason for the inducibility of TE activity is their normally repressive control by epigenetic mechanisms such as cytosine DNA methylation, which are sensitive to perturbations [7,9,14]. Thus, there must be a close relationship between transposon activation and DNA methylation. The enrichment of genes affected by transposons observed in Z9 is similar to the genes affected by methylation reported by other researchers [14], which supports the preceding conclusion.

Considering the characteristics of the spaceflight environment and the proven inducibility of TEs, we speculate that some TEs are activated in the space environment, resulting in a suite of changes in the genome. We used space mutation of T75 to obtain a new variety (Z9) with several characteristics optimized. In this study, we conducted a detailed comparative annotation and analysis of the two genomes to investigate the effects of spaceflight manifested in Z9 compared with the wild type T75.

## 2. Results

### 2.1. Agronomic Trait Differences between T75 and Z9

Significant differences in the number of branches (BN), number of pods per plant (PN), number of nodes in the main stem (NN), lowermost pod height (BPH), plant height (PH), number of seeds per plant (SN), seed weight per plant (SWPP), and 100-seed weight (SW) between T75 and Z9 were observed in 2021 (Figure 1). These differences indicated the rich phenotypic diversity of Z9 compared with T75. The data for agronomic characters from two consecutive years were basically consistent (Appendix A). Z9 produced smaller seeds but a greater number of seeds and seed weight per plant compared with T75. Consistent with these results, in our previous research we showed that Z9 exhibits superiority for a suite of agronomic traits, including yield, good eating quality, and resistance to soybean mosaic virus and salt stress, compared with T75.

### 2.2. Whole-Genome Sequencing of T75 and Z9

For data reliability, we collected two samples from each soybean material for whole-genome sequencing. The genomes of T75 and Z9 were sequenced using Illumina sequencing technology, producing 27.62–43.22 Gb of data that covered 28- to 44-fold of the soybean ‘Williams 82’ reference genome (Wm82.a1.v1.1, 914 Mb; hereafter Wm82). The high sequencing depth satisfied the requirement for reliable detection of structural variations (SVs). We mapped all clean reads to the Wm82 reference genome. The proportion of total reads that were mapped attained 98.39–98.77% (Table 1), indicating that the selection of the reference genome in this experiment was appropriate and the data met the requirements for further analysis.

### 2.3. Structural Variations between T75 and Z9

To understand the genomic differences and commonalities between the two sample groups (T75 and Z9) and Williams 82 (the most commonly used soybean reference genome), we compared the T75 and Z9 genomes to each other and with the Wm82 reference genome for single-nucleotide polymorphisms (SNPs), small insertions/deletions (InDels, length < 70 bp) and SVs, which included translocation breakends (BND), deletions (DEL), duplications (DUP), insertions (INS), inversions (INV), and copy number variation (CNV).

The total number of SNPs and InDels between T75 and Z9 was 747,072, compared with 1,615,802 or 1,609,221 between T75 and Wm82, and 1,743,320 and 1,788,615 between Z9 and Wm82 (Table 2). These results confirmed the similarity of the genetic background between T75 and Z9 and the strong repeatability between the two replicates. The distributions of SNPs and InDels between T75 and Z9 varied on the chromosomes, which were mostly consistent with gene density (Figure 2).

We used the software Manta (1.4.0) and CNVnator to detect SVs between T75 and Z9. The SVs are polymorphisms that are known to impact genome composition at the interspecific level and are associated with phenotypic variation. The numbers of BND, DEL, DUP, INS, INV, and CNV between T75 and Wm82, Z9 and Wm82, and T75 and Z9 are shown in Table 3. The total number of SVs between T75 and Z9 was 12,028 (Appendix A), compared with 41,037 or 39,806 between T75 and Wm82, and 40,578 or 44,269 between Z9 and Wm82. The most frequent SV types detected were BND, DEL, and INS between T75 and Z9.

The distribution of SVs between T75 and Z9 varied on the chromosomes. Some were mostly consistent with gene density, as in chromosomes 4, 5, 6, 8, 11, 14, 16, and 20 (Figure 2). We validated nine SVs using PCR, most of which were confirmed (Figure 3).

The aforementioned comparisons also indicated that Z9 was more closely related to T75 than to Wm82. Many variations of Z9 may occur owing to its spatial induction from T75.

### 2.4. Discovery of Transposon Alterations between T75 and Z9

To determine the difference in TE content between T75 and Z9, we compared the 12,028 SVs (the differential SVs between T75 and Z9) with the SoyTE transposon database (https://www.soybase.org/soytedb/) using the alignment software BWA (v0.7.12).

The total number of differential TEs between T75 and Z9 was 29,063 (Appendix A). The distribution of differential TE types on each chromosome differed (Figure 2 and Figure 4). The left part of Figure 4 shows the total differential TE distribution on each chromosome and indicates that TE variations were greatest in chromosomes 16, 18, and 19, whereas the smallest were in chromosomes 5, 11, 15, and 17. We then compared each chromosome at the TE subfamily level. The abundance of most TE subfamilies was similar to the total TE of each chromosome; that is, the TE variations were unevenly distributed across the 20 chromosomes of the T75 and Z9 genomes. Thus, TEs may show a preference for chromosomal insertion and the TE subfamilies might reflect this preference, and a single subfamily does not show a preference different from other subfamilies. We also observed that the number of retrotransposons (Class I) was much larger than the number of transposons (Class II), which was consistent with the distribution of the two types in plants [17].

### 2.5. Genes Associated with SVs and TE Variations between T75 and Z9

In this study, Manta and CNVnator identified 12,028 potential SVs. We detected 29,063 differential TEs based on the 12,028 SVs. Querying these variant sites in the reference genome, we observed that 1336 potential genes were variable between T75 and Z9 in terms of SVs and TEs. These differential genes were hypothesized to have been mainly acquired through the space mutation of T75. A small proportion of the differential genes might have been caused by manual selection and natural variation in the breeding process.

Comparison of sequencing alignments against the NCBI Non-Redundant Protein (NR), Gene Ontology (GO), and the Kyoto Encyclopedia of Genes and Genomes (KEGG) databases revealed that 1336 differential genes were annotated with a functional description (Appendix A). Of these differential genes, 848 (63.47%) genes were annotated with GO slim terms for plants in the categories of molecular function, cellular component, and biological process (Figure 5A). Analysis of GO term enrichment revealed that variable genes tended to be enriched in functions such as response to stimulus and stress, nitrogen compound metabolic process, reproduction, carbohydrate metabolic process, signal transduction, lipid metabolic process, regulation of biological quality, carbohydrate metabolic process, polysaccharide metabolic process, phosphorylation, growth, glucan metabolic process, cell wall, and methylation (Figure 5B). This was consistent with the phenotypic variation of Z9 in resistance to soybean mosaic virus and salt stress, yield, and seed weight per plant.

## 3. Discussion

The spaceflight environment contains complex mutagenic factors that might activate TEs and cause a series of changes in the genome [16,18]. In plants, the transcriptional activation of TEs has been reported to be induced by a wide array of activating stimuli [15,19]. Transposon activation has caused copy number variation or translocation of a DNA fragment [6,14,16]. We used space mutation of T75 to obtain a new variety (Z9) with several characteristics optimized, including yield, good eating quality, and resistance to soybean mosaic virus and salt stress. The spaceflight environment caused numerous phenotypic mutagenic changes to T75 and, in addition, the offspring of Z9 were more prone to phenotypic variation in actual production. Given that most of the phenotypic variations were heritable, we consider that the phenotypic variation and genetic variation between T75 and Z9 are interrelated. Considering all of these observations, we inferred that TEs might play an important role in the space mutation of T75, even after multiple generations of selective breeding to obtain Z9.

We detected 1336 potential genes that were variable between T75 and Z9 based on 29,063 differential TEs and 12,028 SVs. The variable genes might be caused by space mutation, which is associated with the phenotypic mutagenesis in Z9 compared with T75. The variable genes tended to be enriched in functions such as epigenetics, defense response, cell wall-related processes, auxin metabolism and transport, and signal transduction. Although space breeding experiments of different plants have yielded diverse results owing to a variety of factors, changes in cell wall-related processes and defense responses have been consistently detected [5]. The present results at a genetic level were consistent with this finding.

The cellular epigenetic maintenance machinery of certain rice genotypes is susceptible to perturbation under the spaceflight environment, which results in not only epigenetic changes per se but also de-repression and transposition of certain potentially active TEs, and these cause further genetic changes [6]. The genetic changes in epigenetic-related genes might conversely regulate epigenetics and TE activity. This suggests that the epigenetic maintenance machinery and TE activity might constitute a regulatory loop. Twenty-eight potential genes involved in methylation were variable between T75 and Z9. Four of these genes (Glyma13g05430, Glyma13g38290, Glyma19g13480, and Glyma19g17460) encode histone lysine methyltransferases, which direct H3K4 (an extensively reported epigenetic marker for transcriptional initiation) methylation in Arabidopsis [20,21,22]. Extensive previous studies have shown that histone lysine methylation plays a critical role in epigenetically regulating various biological processes, including fiber cell wall development [23]. These results support the inference that the genetic changes caused by TE activity in turn affect epigenetics. We also observed similar gene enrichment in epigenetic changes of Arabidopsis seedlings under spaceflight [14] and TE alterations of Z9 mutated by spaceflight. Thus, spaceflight is strongly associated with epigenetic regulation and transposon activation. Below, the enriched GO terms of variable genes between T75 and Z9, which are similar to gene enrichment in epigenetic changes of Arabidopsis seedlings under spaceflight, are discussed in detail in the following sections.

It is reasonable to consider that the defense response is stimulated in plants during spaceflight. To survive in stressful environments, such as space, TEs might promote genomic restructuring and plasticity to generate genetic, biochemical, and phenotypic novelties to adapt to external changes [24,25]. We detected 278 potential genes involved in stimulus or stress response that were variable between T75 and Z9. This is consistent with the phenotypic variation of Z9. These potential genes might play a role in the enhanced resistance to soybean mosaic virus and salt stress of Z9 compared with that of T75 [3]. The present research is consistent with previous results that TE dynamics are impacted by response to stress and thus may reflect the evolution and adaptation of their host to the environment [10,16,24,25].

The spaceflight environment could alter cell shape and cell wall components as a result of cell wall perturbation and change the expression of genes involved in cell wall structure [5,14]. We observed that 27 potential genes involved in the cell wall were variable between T75 and Z9. Two of these genes (Glyma10g07310 and Glyma07g27450) encode pectinesterase (PE; E.C. 3.1.1.11), which is an enzyme responsible for the demethylation of galacturonyl residues in high-molecular-weight pectin and is considered to play an important role in cell wall metabolism [26]. One of the 27 genes, Glyma13g27250, encodes cellulose synthase, which is an important enzyme for cellulose production in the plasma membrane of plant cells [27,28]. The cell wall is composed of a cellulose–hemicellulose network and pectic polysaccharides, which is essential to maintain the mechanical strength of the cell wall [29]. The plant cell wall rigidity could be altered under microgravity, which is one factor of the spaceflight environment [5,14]. These results show that genetic modification via TEs of the cell wall biosynthetic process is an important step in gravity resistance.

Gravity sensing in plant cells involves auxin metabolism, transport, and signal transduction [14]. The morphogenesis of plant organs could be changed owing to the effects of microgravity on the polar transport of auxin [30]. We determined that 26 potential genes involved in auxin metabolism and auxin transport were variable between T75 and Z9. Seven of these genes encode ABC transporters, which play a role in the transport of the phytohormone auxin [31]. The enrichment of differential genes encoding ABC transporters implies that auxin transport can be modulated in response to spaceflight and may be caused by TE activity. Abundant differential genes (77 of 1336) were enriched in signal transduction, including auxin, ethylene, abscisic acid, salicylic acid, and jasmonic acid. Alterations in the signal transduction genes, especially auxin-related genes, may play important roles in T75 adaptation to the spaceflight environment and the optimized characters in Z9.

This study is the first to evaluate the TE variation in the genome structural changes of soybean under space breeding. Transposon insertion exhibits a certain degree of randomness; that is, it may be conducive to plant adaptation to the environment, or it may not be conducive to plant growth. At an advanced stage of space breeding, we might have eliminated the mutations that are not conducive to plant growth. Therefore, we may have lost a small proportion of the differential genes in the advanced breeding process. However, this will have little impact on the present analysis and understanding of the overall TE variations.

In the initial stage of this experiment, two soybean varieties, T75 and ‘Zhexian 4’, were carried by the recoverable satellite Shijian 8. The latter variety showed no visible variation, whereas T75 exhibited abundant visible variation. Because there was only one sample that was non-sensitive to space mutagenesis, we did not include it in this report. However, we preliminarily analyzed the difference in transposons between Zhexian 4 and T75 and detected transposon alterations in some methylation-related genes, including genes encoding histone-lysine *N*-methyltransferase. Combined with the following points: (1) the progeny of Z9 were more susceptible to phenotypic variation than T75; (2) 28 potential genes involved in methylation were variable between T75 and Z9; and (3) similar gene enrichment was observed between T75 and Z9, and in epigenetic changes of Arabidopsis seedlings under spaceflight. Assuming that transposon insertion exhibits a certain randomness, we conclude that DNA methylation is closely associated with TE activation and is likely to interact with each other. Previous reports have shown that certain genotypes are preferentially susceptible to space mutation. Therefore, we infer that transposons at certain sites are susceptible to environmental mutagenesis and operate in conjunction with methylation to adapt to environmental changes. Of course, more samples are needed to confirm this hypothesis and apply it to a selection of aerospace breeding materials. We consider that these issues warrant further investigation.

## 4. Materials and Methods

### 4.1. Plant Materials and Field Experiments

In this study, the soybean cultivars ‘Taiwan 75’ (T75) and ‘Zhexian 9’ (Z9) were used. T75 was carried by the recoverable satellite Shijian 8 for the space mutation treatment. Z9 was selected by a pedigree breeding method from among the variable material of T75. Z9 was selected by the Nuclear Institute of the Zhejiang Academy of Agricultural Sciences.

Field trials were conducted at the experimental farm of the Zhejiang Academy of Agricultural Sciences (30°30′76.63″ N, 120°19′55.16″ E) from March to July in 2021 and 2022. The two cultivars of soybean were grown in a random block design with three replications. A plot size of 3 m × 1 m was used, with spacing of 0.2 m and 0.1 m between rows and individual plants, respectively. Data for eight agronomic traits were collected. Plant height (PH), lowermost pod height (BPH), number of branches (BN), number of nodes in the main stem (NN), number of pods per plant (PN), number of seeds per plant (SN), seed weight per plant (SWPP), and 100-seed weight (SW) were measured. Data for seven agronomic traits, excluding SW, were collected from 20 randomly selected plants in the three plots. Data for SW was collected on a whole-plot basis with three replications. The data were analyzed with EXCEL 2013, LSD Multiple Comparison in ANOVA and box plots were generated using the OmicShare tools, a free online platform for data analysis (https://www.omicshare.com/tools, accessed on 7 August 2015). A Student’s *t*-test was used to analyze the significance of differences between the means for the two varieties.

### 4.2. DNA Sequencing of T75 and Z9

Leaves of T75 and Z9 were sampled 15 days after sowing seeds. Genomic DNA was extracted. The DNA libraries were constructed using the Illumina TruSeq DNA Sample Prep Kit and sequenced on an Illumina sequencing platform by Genedenovo Biotechnology Co., Ltd. (Guangzhou, China). The raw image data obtained by sequencing were transformed into sequence data by base calling to generate the raw reads. Low-quality reads (Q ≤ 20), adapter sequences, and reads with N > 10% were removed from the dataset with FASTP [32]. The filtered reads were aligned to the Wm82 reference genome using the MEM algorithm implemented in the BWA alignment software [33]. After comparison, the Picard (v1.129) software (http://sourceforge.net/projects/picard/, accessed on 7 August 2015) was used to mark the repeated reads, and the depth and coverage of the marked reads were counted.

### 4.3. Analysis of SNPs and InDels

Detection of SNPs and InDels (length < 50 bp) was performed using the variant detection software GATK. The detected variants were functionally annotated using ANNOVAR [34].

### 4.4. Identification of Structural Variations

Structural variations were large-scale chromosomal structurally variable regions in the genome, including deletions (DEL), duplications (DUP), insertions (INS), inversions (INV), and copy number variation (CNV, length > 1 kb). We used Manta (1.4.0) [35] and CNVnator (0.3.2) [36] to detect large structural variants and copy number variation.

### 4.5. Transposable Element Analysis

A method similar to that for SV annotation was adopted to determine TEs. First, SVs were annotated according to the reads with inconsistent alignment to locate the positions where transposons may occur. The sequences neighboring these positions were compared with sequences in the SoyTE transposon database (https://www.soybase.org/soytedb/) [17] to determine if transposons were locatable near these positions using the BWA alignment software.

### 4.6. Identification of SVs and TE-Related Genes, and PCR Validation

Structural variation-related and TE-related genes were identified by screening the annotated genes that contained or were adjacent to differential TEs. The gene sequences were aligned with sequences in the NR (NCBI non-redundant protein), Swiss-Prot, GO, COG (Clusters of Orthologous Groups), and KEGG databases using BLAST2GO (v5.2.5, https://www.blast2go.com/).

Nine SVs were validated by PCR amplification of the sequences of T75 and Z9. Gene-specific primers were designed using the Primer-Blast tool available on the NCBI website (https://www.ncbi.nlm.nih.gov/tools/primer-blast/). All primers are listed in Table 4.

## Figures and Tables

**Figure 1 ijms-23-13721-f001:**
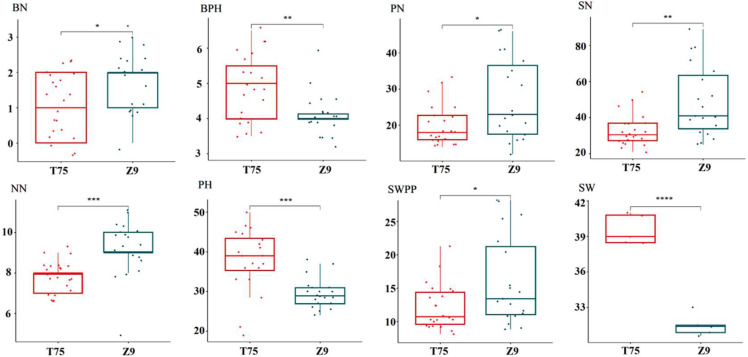
Agronomic traits of soybean ‘Taiwan 75’ (T75) and ‘Zhexian 9’ (Z9). BN, number of branches; PN, number of pods per plant; NN, number of nodes in main stem; BPH, lowermost pod height; PH, plant height; SN, number of seeds per plant; SWPP, seed weight per plant; and SW, 100-seed weight. * *p* < 0.05, ** *p* < 0.01, *** and **** *p* < 0.001 (Student’s *t*-test).

**Figure 2 ijms-23-13721-f002:**
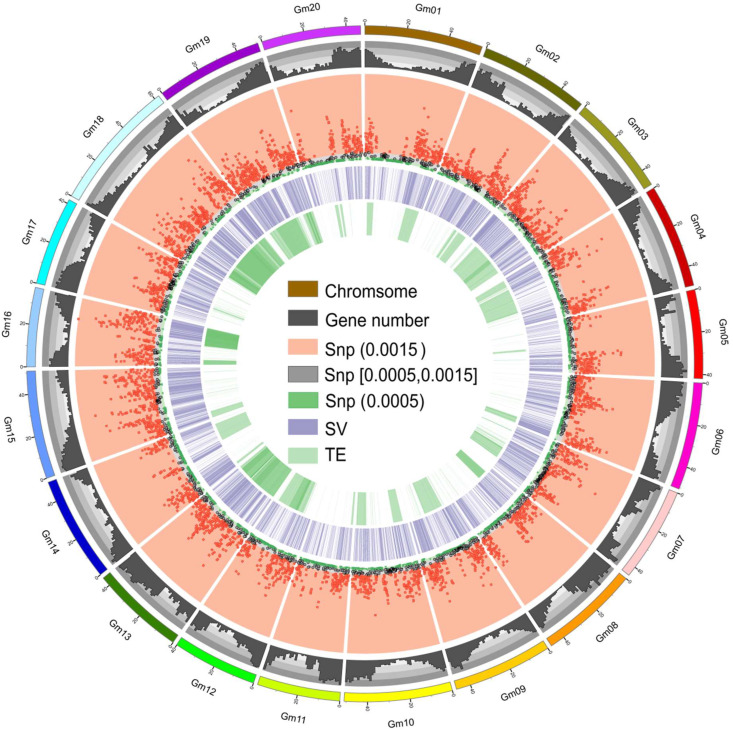
Circos plot showing differences between the soybean ‘Taiwan 75’ (T75) and ‘Zhexian 9’ (Z9) genomes. Tracks from outer to inner circles indicate: chromosome, gene number in reference genome, single-nucleotide polymorphisms (SNP) and small insertions/deletions (InDels), structural variations (SV), and transposable element (TE) content. The window size is 100 kb, and 0.0015 and 0.0005 are the SNP densities of the window.

**Figure 3 ijms-23-13721-f003:**
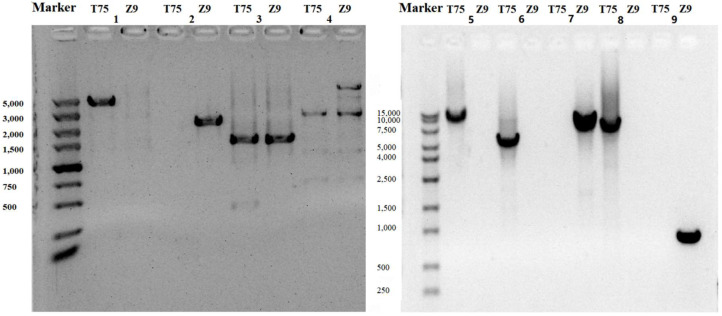
Verification by PCR of nine structural variations in the soybean ‘Taiwan 75’ (T75) and ‘Zhexian 9’ (Z9) genomes.

**Figure 4 ijms-23-13721-f004:**
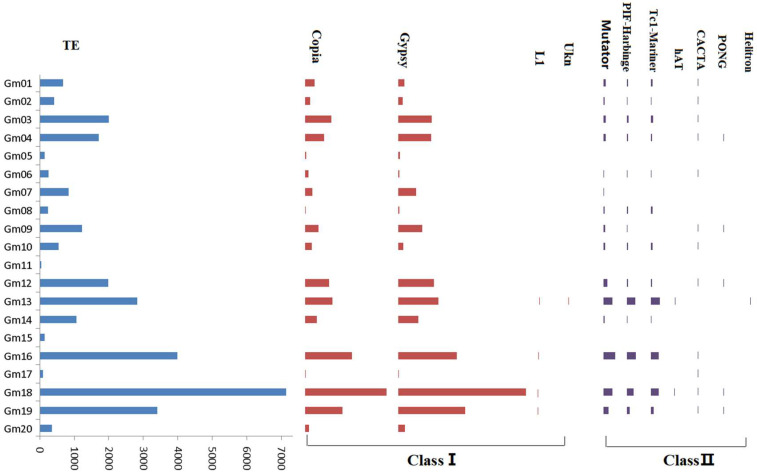
Distribution of differential transposable element (TE) superfamilies in each soybean chromosome. The *x*-axis represents the number of TEs.

**Figure 5 ijms-23-13721-f005:**
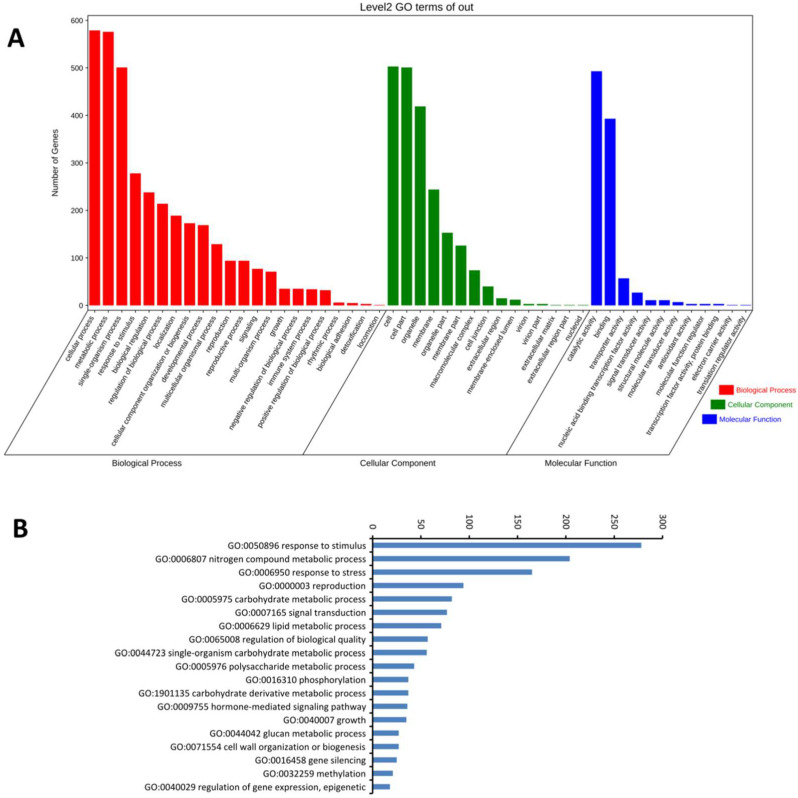
Gene ontology (GO) classification of the differential genes altered by transposable elements. (**A**) The differential genes were assigned to three GO categories (cellular component, molecular function, and biological process). (**B**) GO-biological process enrichment of the differential genes. The bar length represents the number of genes in the test set belonging to each GO category.

**Table 1 ijms-23-13721-t001:** Comparison of genome statistics for each sample of soybean ‘Taiwan 75’ (T75) and ‘Zhexian 9’ (Z9).

Sample ID	Clean_Data (Gb)	Total Read	Total UnMapped	Total Mapped
T75-1	35.64	253,778,974	1.44%	98.56%
T75-2	27.62	196,657,892	1.61%	98.39%
Z9-1	28.06	199,679,784	1.39%	98.61%
Z9-2	43.22	307,682,454	1.23%	98.77%

**Table 2 ijms-23-13721-t002:** Genomic location of single-nucleotide polymorphisms (SNPs) and small insertions/deletions (InDels) in soybean ‘Taiwan 75’ (T75) and ‘Zhexian 9’ (Z9). Downstream, the variant overlaps a 1 kb region downstream of the transcription end site; an exonic, variant overlaps a coding exon; intergenic, variant is in an intergenic region; intronic, variant overlaps an intron; splicing, the variant is within 2 bp of a splicing junction; upstream, variant overlaps a 1 kb region upstream of the transcription start site; UTR5 and UTR3, variant overlaps a 5′ or 3′ untranslated region, respectively.

Sample	Downstream	Exonic	Exonic; Splicing	Intergenic	Intronic	Splicing	Upstream	Upstream; Downstream	UTR3	UTR5	UTR5; UTR3	Total
T75-1	98,838	66,681	17	1,160,059	150,634	704	109,347	9302	13,749	6402	69	1,615,802
T75-2	98,393	66,482	17	1,155,317	150,316	701	108,614	9201	13,713	6394	73	1,609,221
Z9-1	108,812	74,277	18	1,239,329	167,829	796	119,011	10,310	15,678	7157	103	1,743,320
Z9-2	112,418	74,665	18	1,272,167	169,955	802	124,783	10,682	15,810	7204	111	1,788,615
T75 vs. Z9	52,826	31,016	8	507,051	82,154	373	56,340	5281	8172	3792	59	747,072

**Table 3 ijms-23-13721-t003:** Structural variation types between soybean ‘Taiwan 75’ (T75) and ‘Zhexian 9’ (Z9) genomes.

Sample	BND	DEL	DUP	INS	INV	CNS	Total
T75-1	22,468	7620	1049	2596	845	6459	41,037
T75-2	21,930	7185	1032	2432	821	6406	39,806
Z9-1	21,578	7720	1004	2844	880	6552	40,578
Z9-2	23,676	8718	1092	3185	944	6654	44,269
T75 vs. Z9	6428	3097	307	1332	295	569	12,028

**Table 4 ijms-23-13721-t004:** Primers designed for validation of nine structural variations.

Primer	Gene Name	Product Size (bp)	Sequence (5′-3′)
G1-F	Glyma16g22390	4399	GCCAAAGGGAAGCTTGGAGA
G1-R			GGCGTCCTACATGTTGCCTA
G2-F	Glyma02g38740	2313	CGCTAGCTCTGCGATCATGT
G2-R			TTGTACCGCTGCTGAGAACA
G3-F	Glyma07g19820	1509	TCAGACGAGCTGTACCCATC
G3-R			CATGTGTTTCGCCCTTGTGG
G4-F	Glyma07g13230	7711	TTGTTGAGACGCGCTTTGTG
G4-R			TTGCTGGAAGGAGCCAAGAG
G5-F	Glyma06g02600	9392	ACAATCACCGTTCGTCGGAA
G5-R			AGAACGGAAGTGAACGCAGA
G6-F	Glyma07g26460	5019	AGTGGCACATCCGAAGTGAG
G6-R			ATGGTTTTCTTGGTGGGGCA
G7-F	Glyma02g02860	6462	TTGCTCCGTGTTTGACCTGT
G7-R			TCAAACCAGGGTGCTTCGTT
G8-F	Glyma15g26810	6338	TGGCAACTACGCACAATCCT
G8-R			ACCACCACATTCCTCGTGAC
G9-F	Glyma18g07740	757	GGCCACGATGTGGAAGAGAC
G9-R			TGCATAGCCCTCCACATTCC

## Data Availability

The datasets generated and/or analyzed during the current study are available from the corresponding author on reasonable request.

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
