# Peer review of "Genome-Wide Comparison of Structural Variations and Transposon Alterations in Soybean Cultivars Induced by Spaceflight"

_ijms, 2022, doi:10.3390/ijms232213721_

Round 1

Reviewer 1 Report

I find that this manuscript has major drawbacks. Please see my comment below.

English language and syntax needs improvement throughout the text. Many sentences do not read well and need major revision.

In the Abstract, please state clearly the objective of your research study, the reasons you undertook this project, and the benefits of your results for the plant breeders.

Introduction is very short and needs revision. Please add more information and literature on the space mutation treatment as well as the cultivars T75 and Z9.

Data presentation is not focused. Please revise and summarize accordingly.

It is very difficult to judge the merit and scientific soundness of this project because certain important information is missing. There is no information on the experimental design of the phenotypic data.

Authors state that the two soybean genotypes were Taiwan 75 (T75) and Zhexian 9 (Z9).  “Z9 was created by the Nuclear Institute of Zhejiang Academy of Agricultural Sciences, using space mutation breeding from T75”.

The term Space mutation breeding is general. What exactly was the procedure? State the details, how was Z9 developed? How long did it take for the development of Z9? Please add all the relevant details that the breeders would want to know.

Regarding the phenotypic data, there is practically no information. Did you only use ONE year of data?

What was the experimental design? How many replications? How many rows? What was the row spacing and plant spacing?

When was the trial planted and harvested?

Authors conclude that “the results of this analysis are beneficial for understanding the mechanism of plant adaptation to spaceflight and improve strategies to allow plants to adapt to space.”

I do not understand how this would help plant breeders practically. I do not see how these results are beneficial to the plant breeders or agronomists. Please elaborate. How would you use plant adaptation to spaceflight to raise crop yield and stability of performance?

Consider presenting this as a Note type of paper, not an Article

Author Response

I find that this manuscript has major drawbacks. Please see my comment below.

1.English language and syntax needs improvement throughout the text. Many sentences do not read well and need major revision.

We have entrusted a professional English editing company Liwen Bianji (Edanz) (www.liwenbianji.cn) for editing the English text.

  1. In the Abstract, please state clearly the objective of your research study, the reasons you undertook this project, and the benefits of your results for the plant breeders.

We have modified Abstract the according to the comments.

3.Introduction is very short and needs revision. Please add more information and literature on the space mutation treatment as well as the cultivars T75 and Z9.

We have added more information and literature on the space mutation treatment and TE at line 43-52, and line 73-81.

4.The term Space mutation breeding is general. What exactly was the procedure? State the details, how was Z9 developed? How long did it take for the development of Z9? Please add all the relevant details that the breeders would want to know.

We have added the details at line 43-51.

5.Regarding the phenotypic data, there is practically no information. Did you only use ONE year of data?

We use two years and have added the two year data at Table S1. The phenotypic data is only a part of the data that we have supplemented to analyze and discuss the phenotypic differences and transposon changes. In fact, in another 2019 Chinese literature, we have conducted a systematic analysis of the agronomic character differences between Z9 and T75, but T-test analysis was not conducted. In this article, we investigated the differences of agronomic traits again, on the one hand, to determine the stability of these two varieties. The results in 2021 and 2022 were basically consistent with those in the 2019 literature, which also proved that although the offspring of Z9 were more likely to separate than T75, the agronomic traits of these two varieties are basically stable.

6.What was the experimental design? How many replications? How many rows? What was the row spacing and plant spacing?When was the trial planted and harvested?

We have added this information in ‘4.1. Plant materials and Field experiments’ part.

7.Authors conclude that “the results of this analysis are beneficial for understanding the mechanism of plant adaptation to spaceflight and improve strategies to allow plants to adapt to space.”

I do not understand how this would help plant breeders practically. I do not see how these results are beneficial to the plant breeders or agronomists. Please elaborate. How would you use plant adaptation to spaceflight to raise crop yield and stability of performance?

We miss some important point in the discussion. Now we added it which will answer the comments.

Reviewer 2 Report

In the article “Genome-wide comparisons of structural variations and transposon alterations in soybean cultivars induced by spaceflight” the authors describe the analysis of the line Z9 produced by “space mutation treatment” of the wild type T75. The pretense of the article is interesting, but in the current state a lot of background information is missing.

Major comments:

1. Space mutation treatment carried by recoverable satellites:  Please describe in more detail, when and how long were the plants (seeds?) in the space mutation treatment. Please cite former articles. Is it a method to mutageneize the seeds (I suppose)?

2. What was the orbit height of the satellite? The authors mention in the introduction a variety of factors that can influence the breeding, and also mention the space magnetic field. As far as my understanding goes, the earth magnetic field is covering most (or all area) where humans operate orbiting objects. Even the ISS is still covered by the magnetic field. Wihtout further knowledge of your setup, it is likely that you used satellites in the lower stable orbits, which are well in the range of the earth magnetic field. This should severely limit the range of possible mutagenic effects of the “space magnetic field” and “cosmic irradiation” reaching your samples.

3. Was it a one time treatment, or did you send several generations of seeds/plants?

4. For the actual selection of your new “line”, how often did you backcross it to reach a homogenicity. (How often did you backcross it to generate a stable line).

5. In the 3rd Paragraph of your discussion, you describe your results like they come from an RNAseq. As far as I know these kind of mutations is fairly random, and only the selection process would enhance the factors that might be important for particular resistance. At least that is the common conception for plant breeding. You indicate in the next paragraph that epigenetic changes might influence the TE structure. This is a feasible argument, but when you want to argue for this purpose, you should set it before paragraph 3. When this is not an argument for paragraph 3, I might misunderstand your intention of paragraph 3. Please specify.

Materials and Methods:

1. Space mutation breeding conditions are not properly described.

2. You write about 2 field trials, data not shown.

Minor comments:

Figure 1: Low picture quality, please increase the font size.  What is the repetition number for all samples? Some of your bars do not indicate a range (no outliers?)

Table 2: I am not sure what is the meaning of the columns: Upstream;downstream and UTR5;UTR3.

In chapter 2.4 Discovery of transposon alterations between T75 and Z9, you have a wrong figure indication. Please check the text and the corresponding figures.

Discussion: It sounds like you already presented a study that describes your new line Z9. Please indicate it by citation or make clear that these data are not provided to the public.

Author Response

Major comments:

  1. Space mutation treatment carried by recoverable satellites:  Please describe in more detail, when and how long were the plants (seeds?) in the space mutation treatment. Please cite former articles. Is it a method to mutageneize the seeds (I suppose)?

Yes, it’s a method to mutageneize the seeds. We have added more information and literature on the space mutation treatment at line 36-44.

  1. What was the orbit height of the satellite? The authors mention in the introduction a variety of factors that can influence the breeding, and also mention the space magnetic field. As far as my understanding goes, the earth magnetic field is covering most (or all area) where humans operate orbiting objects. Even the ISS is still covered by the magnetic field. Wihtout further knowledge of your setup, it is likely that you used satellites in the lower stable orbits, which are well in the range of the earth magnetic field. This should severely limit the range of possible mutagenic effects of the “space magnetic field” and “cosmic irradiation” reaching your samples.

We have added more information and literature of the satellite and space treatment at line 36-44.Other reports proved that the seeds inside the satellite module have received sufficient cosmic irradiation, microgravity, etc.

  1. Was it a one time treatment, or did you send several generations of seeds/plants?

Yes, it was a one-time treatment.

  1. For the actual selection of your new “line”, how often did you backcross it to reach a homogenicity. (How often did you backcross it to generate a stable line).

No, we didn’t backcross it. The T75 seeds were mutated by space environment. Then the new fresh soybean variety Z9 was selected by pedigree method for 6 years.  

  1. In the 3rdParagraph of your discussion, you describe your results like they come from an RNAseq. As far as I know these kind of mutations is fairly random, and only the selection process would enhance the factors that might be important for particular resistance. At least that is the common conception for plant breeding. You indicate in the next paragraph that epigenetic changes might influence the TE structure. This is a feasible argument, but when you want to argue for this purpose, you should set it before paragraph 3. When this is not an argument for paragraph 3, I might misunderstand your intention of paragraph 3. Please specify.

Materials and Methods:

  1. Space mutation breeding conditions are not properly described.

We have modified it according to the comments, and added the detail in the introduction part at line 37-46.

  1. You write about 2 field trials, data not shown.

We have added the two year data at Table S1.

  1. Figure 1: Low picture quality, please increase the font size. What is the repetition number for all samples? Some of your bars do not indicate a range (no outliers?)

We have increased the font size of Figure 1. Data for 7 agronomic traits, except SW, were collected from the 20 randomly selected plants of three plots. Whereas data on SW was collected on plot basis with three replications, and the bars of SW were more concentrated. Some data of agronomic traits might be more stable, without outliers.

  1. Table 2: I am not sure what is the meaning of the columns: Upstream;downstream and UTR5;UTR3.

We have added the meaning of the columns in table 2 notes.

10.In chapter 2.4 Discovery of transposon alterations between T75 and Z9, you have a wrong figure indication. Please check the text and the corresponding figures.

We have modified it and check again.

11.Discussion: It sounds like you already presented a study that describes your new line Z9. Please indicate it by citation or make clear that these data are not provided to the public.

We have modified it according to the comments.